# Dynamic microvilli sculpt bristles at nanometric scale

Kyojiro N. Ikeda [1] ✉, Ilya Belevich [2], Luis Zelaya-Lainez[3], Lukas Orel [1], Josef Füssl[3], Jaromír Gumulec [4], Christian Hellmich[3], Eija Jokitalo [2] & Florian Raible [1,5] ✉

Organisms generate shapes across size scales. Whereas patterning and morphogenesis of macroscopic tissues has been extensively studied, the principles underlying the formation of micrometric and submicrometric structures remain largely enigmatic. Individual cells of polychaete annelids, so-called chaetoblasts, are associated with the generation of chitinous bristles of highly stereotypic geometry. Here we show that bristle formation requires a chitin-producing enzyme specifically expressed in the chaetoblasts. Chaetoblasts exhibit dynamic cell surfaces with stereotypical patterns of actin-rich microvilli. These microvilli can be matched with internal and external structures of bristles reconstructed from serial block-face electron micrographs. Individual chitin teeth are deposited by microvilli in an extension-disassembly cycle resembling a biological 3D printer. Consistently, pharmacological interference with actin dynamics leads to defects in tooth formation. Our study reveals that both material and shape of bristles are encoded by the same cell, and that microvilli play a role in micro- to submicrometric sculpting of biomaterials.

The question of how organisms generate form and patterns is one of the central problems in developmental biology[1,2]. So far, research into this question has mainly focused on the level of macroscopic morphogenesis, addressing how multicellular tissues generate shape on the size scale of milli- to centimetres. However, cells not only participate in such macroscopic morphogenetic movements but can also adopt particular geometries on their own at size scales of micro- to nanometres[3]. The generation of such patterns cannot be explained by the inter-cellular signalling systems that have been recognised to orchestrate macroscopic morphogenesis. This raises the question of which molecular mechanisms are responsible for this fundamentally distinct type of morphogenesis.

Chitin-containing bristles (chaetae or setae) are name-giving features of polychaete worms, but also occur in various related invertebrate taxa. In contrast to insect bristles, which are chitin-coated cells, polychaete bristles are products of basal apposition mechanism,

where more distal parts are synthesised first and subsequently pushed out during synthesis of more basal structures (Fig. 1a). This process yields highly stereotypical geometries, ranging from feather-like structures or hooks to composite structures containing miniature joints (see refs. 4,5. for an extensive review, including ultrastructural work in various polychaete genera). Stereotypicity of a given bristle type within a species indicates the existence of a genetically encoded programme orchestrating bristle biogenesis. Building on fundamental light microscopy work on the *Nereis* genus that reaches back more than a century[6], electron-microscopic work in the nereidid polychaete *Nereis vexillosa* revealed specific cells (chaetoblasts) at the base of each bristle that exhibit varying surface geometries matching the synthesised structure[7]. Due to the lack of a suitable experimental model, however, it has never been tested if the chaetoblast surface is instructive for bristle shape. Likewise, it has remained unclear if the chaetoblast just provides a 3-dimensional mould for extracellular

[1]Max Perutz Labs; University of Vienna, 1030 Vienna, Austria. [2]Institute of Biotechnology, Helsinki Institute of Life Science, University of Helsinki, Helsinki, Finland. [3]Institute for Mechanics of Materials and Structures, TU Wien-Vienna University of Technology, Vienna, Austria. [4]Department of Pathophysiology, Faculty of Medicine, Masaryk University, Brno, Czech Republic. [5]Research Platform "Single-Cell Regulation of Stem Cells", University of Vienna, Vienna, Austria. ✉e-mail: kyojiro.ikeda@univie.ac.at; florian.raible@univie.ac.at

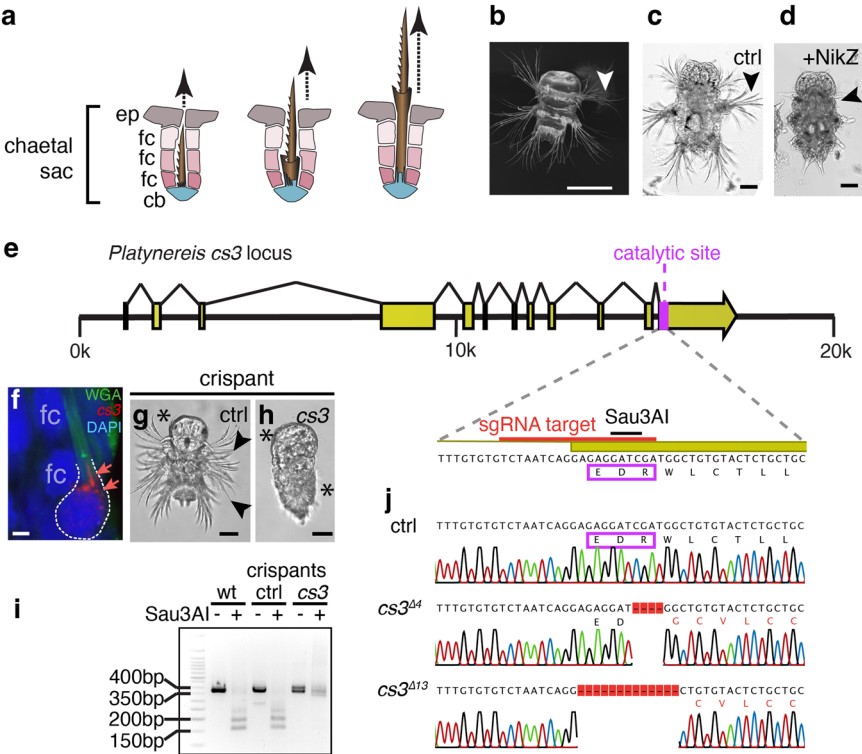

**Fig. 1 | Bristles arise by basal apposition, requiring the activity of a chaetoblast-specific chitin synthase. a** Schemes illustrating the basal apposition principle leading to the emergence of bristles from chaetal sacs comprised of the basal most chaetoblast (cb), follicle cells (fc), and surface epiblasts (ep). **b** Compound bristles (arrowhead) line the sides of larvae at 3 days post fertilisation (3dpf). **c, d** Inhibition of chitin synthase activity by nikkomycin Z (**d**) abrogates bristle formation when compared to control larvae (**c**). **e** Genomic structure of the *chitin synthase 3 / cs3* gene of *Platynereis*, indicating the position of the sequence encoding the catalytic

site targeted in (**h–j**). **f** Probes against *cs3* mRNA (detected in red, arrows) specifically localise to the chaetoblast (cb); chitin co-visualised in green using wheat-germ agglutinate; **g, h** Microinjection of *cas9* mRNA and sgRNA targeting *cs3* yields bristle-less larvae (**h**), unlike microinjections targeting an unrelated gene (**g**). **i, j** Validation of *cs3* lesions in (**h**) by genomic amplification and test for the absence of the endogenous Sau3AI restriction site in wild-type sequence (**i**) and sequencing of amplicons (**j**). Scale bars: **b** 100 µm, **c, d, g, h** 50 µm, **f** 2 µm. Asterisks in **h** demarcate the position of ciliary bands, which appear unaffected.

material secreted from the neighbouring follicle cells—a long-standing model[7–9] that we refer to as a mould/cast mechanism; or if, alternatively, chaetoblasts themselves actively secrete bristle material, in a biological mechanism resembling 3D printing[10]. To solve these fundamental questions, we explored the bristleworm *Platynereis dumerilii*, a laboratory model amenable to molecular experimentation[11].

## Results

### Bristle biogenesis requires a chaetoblast-specific chitin synthase

The early larvae of *P. dumerilii* exhibit a highly stereotypical set of compound bristles that are also referred to as spiniger-type bristles. Bristles of this type are thought to be relevant for locomotion through different substrates[5,12,13]. These bristles arise in specialised tissue pouches—chaetal sacs (schematised in Fig. 1a)—that line the lateral aspects of the larvae (Fig. 1b). To assess if chitin synthase activity was required for larval bristle formation, we incubated larvae prior to bristle formation in 10 µM nikkomycin Z, a competitive inhibitor of chitin synthases used across phyla[14,15]. In contrast to solvent-treated controls (Fig. 1c), larvae in which chitin synthase activity was blocked from 24 to 72 h post fertilisation (hpf) lacked morphologically visible bristles (Fig. 1d). These results are consistent with the notion that chitin is an essential component of bristle formation. However, they do not distinguish between the aforementioned conflicting hypotheses that chitin could either be produced by neighbouring follicle cells, or by the chaetoblasts themselves. To address this question by a genetic approach, we capitalised on prior work in our lab that had identified a specific chitin synthase gene (*cs3*) expressed in chaetal sacs[16]. Analysis

of the gene and its genomic locus (Fig. 1e) confirmed that the encoded Cs3 protein contained an EDR motif (boxed in Fig. 1e) that is part of a diagnostic, evolutionarily conserved chitin synthase signature[17,18]. Consistently, the central aspartate of the motif is known to act as a catalytic residue in the recently characterised chitin synthase structure of a root oomycete, *Phytophthora sojae*[19]. To assess the expression of *cs3* transcripts, we designed *cs3*-specific probes for in situ hybridisation chain reaction (in situ HCR)[20]. Our staining revealed specific expression of *cs3* in chaetoblasts, but not in follicle cells (Fig. 1f). To test if this chaetoblast-specific chitin synthase was required for bristle formation, we turned to Cas9/CRISPR-mediated knock-out technology, which has previously been shown to work in *Platynereis*[21]. We designed a single guide RNA targeting the region encoding the EDR motif, which also contained an endogenous Sau3AI restriction site for probing genomic integrity (schematised in Fig. 1e). In contrast to a sgRNA targeting the unrelated *cwo* gene (ctrl in Fig. 1g), co-injection of the *cs3* sgRNA with *cas9* mRNA yielded embryos lacking visible bristles (Fig. 1h). Genomic amplification from wild-type specimens, *cwo* crispants (controls) and *cs3* crispants allowed us to correlate the bristle-less phenotype with a lack of Sau3AI cleavage (Fig. 1i). Likewise, Sanger sequencing of cloned *cs3* amplicons confirmed the presence of lesioned alleles—abrogating the functional EDR motif—in selected phenotypic *cs3*-targeted specimens, but not control crispants (Fig. 1j). These crispant analyses strongly argue that the chaetoblast-specific chitin synthase Cs3 has a key role for bristle biogenesis. This implies that the chaetoblast not only has a structural role, but itself is involved in chitin biosynthesis.

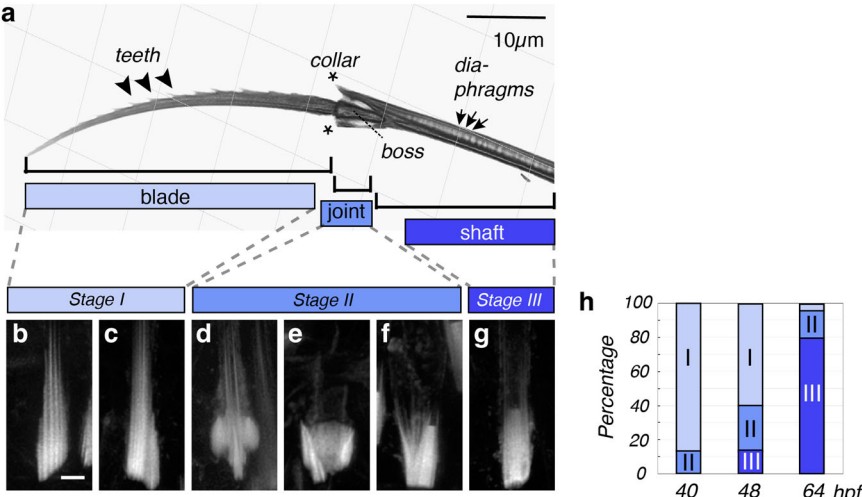

**Fig. 2 | Synchronised, stereotypical changes in chaetoblast microvillar geometry correlate with the production of distinct bristle features. a** Refractive index tomography reveals both micrometric divisions of the bristle (blade, joint, shaft) and repetitive submicrometric features (teeth, diaphragms). **b–g** Correlated geometries of chaetoblast surfaces, as analysed by Phalloidin488 labelling of F-actin in the microvilli of chaetoblasts at synthesis stages I, II and III. Images are Huygen-deconvoluted Z-stack maximal projections of apical chaetoblast surfaces, with distal sides pointing to the top. Scale bar: 2 μm. **h** Distribution of stages represented in (**b–g**) during the development of bristles at 40, 48 and 72 h post fertilisation (hpf). Source data are provided as a Source Data file. Scale bars **a** 10 μm; **b** 2 μm.

## Biogenesis of stereotypical bristle structures is matched by distinct microvillar geometries

To characterise the spatial properties of larval bristles at higher resolution, and deduce the temporal time scales required for their production, we next established a biochemical isolation protocol separating bristles from the surrounding tissue, and employed refractive index tomography on isolated bristles. Tomograms allowed us to visualise diagnostic features of larval bristles on different size scales: they show the tripartite subdivision of the bristle into a micrometre-scale blade, joint and shaft (Fig. 2a and Supplementary Movie 1), and reveal submicrometric features, such as the teeth of the blade, the collar and boss of the joint, and the diaphragms inside the shaft (Fig. 2a). The teeth were consistently aligned on the exterior curvature of the blade. In proximity of the joint, the distance between adjacent teeth measured ~2.5 μm. Towards the tip of the blade, this distance gradually declined, reaching 0.4–1.3 μm at the distal tip. As the formation of a blade requires around 12 h and during that time produces a total of 21 teeth (s.b.), a new tooth is, on average, initiated every ~35 min. In the shaft, diaphragms were regularly spaced at ~0.7 μm distance, indicating that a new diaphragm was deposited every ~12 min. Taken together, our quantification not only supports the idea that the deposition of blades, joints, and shafts is driven by changes in the biogenesis programme on the scale of hours/days, but that additional dynamics on the scale of minutes/hours are responsible for depositing repetitive, submicrometric features of the blade and shaft.

The stereotypicity of bristle structures in *Platynereis*, as well as the documented species-specific variations of bristle patterns in other polychaetes[4,5] indicate that bristle biogenesis is driven by a genetically programmed cellular mechanism. In other contexts, such as the vertebrate hair cells or lateral line neuromasts, elaborate cellular geometries rely on the occurrence of specialised apical microvilli[22]. Consistent with the role of microvilli in polychaete bristle biogenesis, classical ultrastructural work has found cellular protrusions to be associated with bristles in the annelid *Nereis vexillosa*[7], and recent analyses have confirmed largely similar geometries in chaetoblasts of *Platynereis* individuals fixed at different stages of development[9]. To systematically probe for a correlation of microvillar geometry at the cell cortex of *Platynereis* chaetoblasts and the biogenesis of diagnostic bristle features, we performed a systematic analysis of chaetoblast microvilli at distinct times of development associated with the production of characteristic bristle features. As microvilli are usually characterised by filamentous actin (F-actin) fibres, we used the F-actin-binding molecule phalloidin to assess microvillar geometries.

For our analyses, we chose larvae fixed at 40, 48 or 64 hpf, which are developmental stages at which blade/teeth, joint, and shaft are formed, respectively. Phalloidin-based visualisation of actin in ~2400 chaetoblasts by confocal microscopy revealed distinct geometries of microvillar patterns in chaetoblasts that we could assign to the timeline of bristle biogenesis (Fig. 2b, c): At stage I, when the blade with the teeth is forming, we consistently observe a set of ~10 microvilli of different length, collectively forming a cone-shaped assembly (Fig. 2b). This assembly is eventually accompanied by a slim side group of microvilli that are also aligned parallel to each other (Fig. 2c). A diagnostic feature of stage II is the occurrence of a crescent-shaped microvillar arrangement that surrounds a cone formed by parallel microvilli (Fig. d). Progressively, the crescent shape forms a cup-shaped object (Fig. 2e). In a subset of stage II patterns, the cup-shaped object includes a larger, flat-tipped microvillus on one side (Fig. 2f). Stage III is characterised by the presence of a central, flat-tipped axial microvillus of 0.5–0.8 μm diameter (Fig. 2g), flanked by a group of paraxial annular microvilli. The frequency of the different pattern categories by developmental stage (Fig. 2h) not only supports the aforementioned timeline, but also attests to a high degree of synchronicity in the development of individual compound bristles within the developing larva. To test if the formation of microvilli depended on chitin synthesis, we assessed actin geometries after NikZ-mediated inhibition of chitin synthesis and in bristle-less *cs3* crispants. In both cases, microvilli were consistently present, even though the fine morphology of the microvillar assembly exhibited abnormalities (Supplementary Figs. 1, 2). Taken together, our analyses support the notion that stereotypic arrangements of a dynamic microvillar programme are correlated with individual steps of *Platynereis* bristle formation.

## A mechanistic model for microvillar-based printing of submicrometric bristle features

We next assessed to which degree the presence of individual microvilli could be traced to fine details of the produced bristle. Light-microscopic

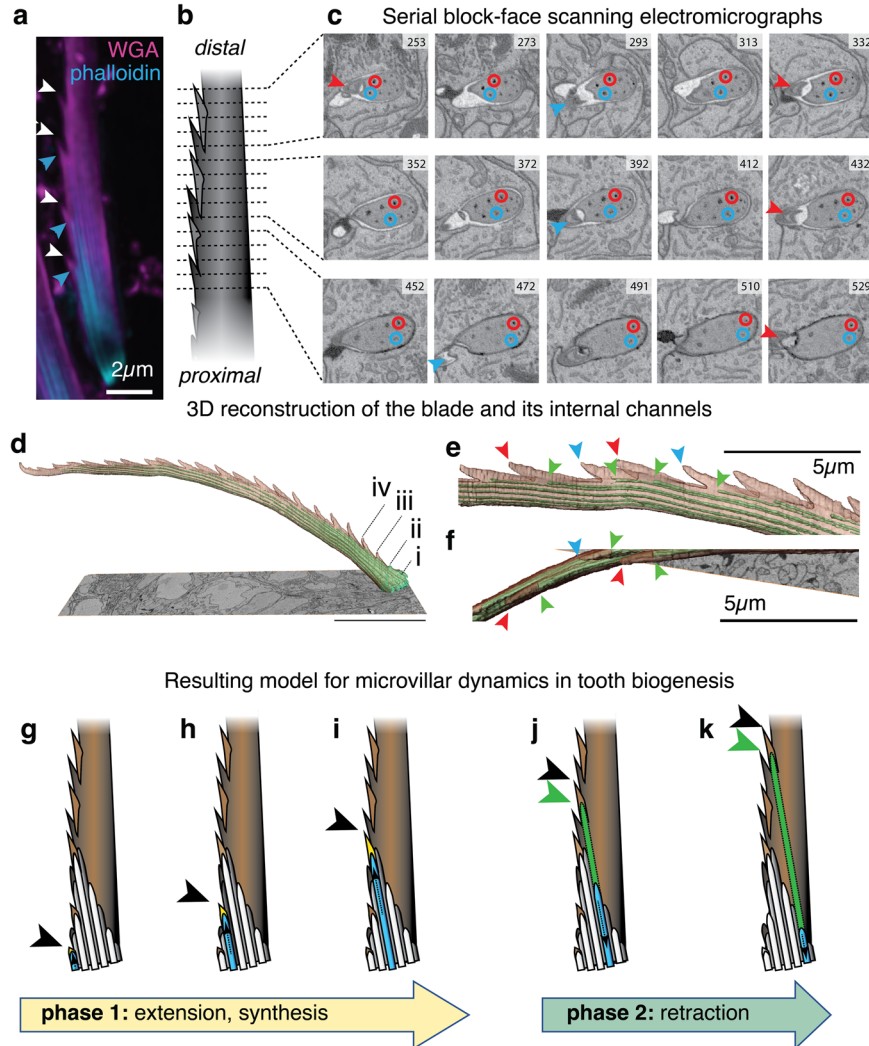

**Fig. 3 | Individual microvillar dynamics explain tooth biogenesis. a** Co-visualisation of chitin (WGA) and F-actin (phalloidin) suggests a correlation of individual microvilli (blue arrowheads) with individual teeth (red arrowheads); **b–f** SBF-SEM analysis; **b** Overview scheme and **c** selected (numbered) ~40 nm cross sections obtained by SBF-SEM on a synthesised blade (see Supplementary Movie 2). Arrowheads demarcate the chitin tip of left (red) or right (blue) teeth appearing prior to the occurrence of the corresponding channel. Red and blue circles trace two of the channels across the represented stack, revealing a systematic displacement towards the non-serrated edge, where older channels are discontinued. **d–f** Overview and details of the resulting 3D reconstruction (cf. Supplementary Movie 3) of the reconstructed blade; red and blue arrowheads demarcate tooth tips, green arrowheads demarcate tips of corresponding channels; in (**d**), (i–iv) demarcate the first four teeth that exhibit an increase in length. **g–k** Resulting model for the synthesis of a given tooth, fitting the triangle-shaped arrangement of microvilli (dark/light grey) observed in Fig. 2b, c into the base of the bristle. The model predicts that extension (phase 1) is associated with active chitin biosynthesis/deposition (yellow); subsequent disassembly of actin (phase 2) would leave the characteristic channels (schematised in green). Arrowheads demarcate tooth tips (black) and tips of channels (green). The direction of bristle growth is indicated by brown, dashed arrows.

analyses suggested that individual microvilli in stage I arrangements could be associated with the formation of individual teeth (Fig. 3a). To examine the details of this potential correlation in more detail, and gain insight into the question how a rather constant set of ~10 microvilli produces twice as many teeth, we turned to serial block-face scanning electron microscopy (SBF-SEM), a technique able to reconstruct micrometric objects from serial electron micrographs sampled at nanometric distance[23]. For analysing the relationship between teeth and microvillar arrangements, we investigated 32 chaetoblast samples at 48 hpf, when blade production should be mostly completed. Machine-assisted modelling of more than 1000~40 nm slices of a representative chaetoblast (Fig. 3b, c and Supplementary Movie 2), followed by manual polishing, allowed us to reconstruct a 46.4 μm blade with 21 teeth (Fig. 3d–f and Supplementary Movie 3). The inside of the SBF-SEM-based blade model is characterised by longitudinal channels. According to early work in nereidid polychaetes[6,7], only channels close to the

chaetoblast contain the microvilli observed in the aforementioned analysis, whereas the distal channels result from the disassembly of microvilli after the solidification of the chitin matrix (Fig. 3c and Supplementary Movie 2). Tracking these channels across the distal-proximal axis revealed that they were neither continuous nor exactly aligned with the main axis of the blade. Rather, individual channels had lengths of around 10–20 μm. Their distal-most tips were positioned close to the serrated edge, while they ended close to the opposite edge of the blade (see circles in Fig. 3c). Spacing and size of teeth in the SBF-SEM-based reconstruction matched our light-microscopic observations. Each tooth was accompanied by a channel that started at a mean distance of ~1.6 μm from the tooth tip (Fig. 3e). The orientation of teeth at the outer edge of the blade alternated in a regular fashion, pointing either to the left or the right face of the blade (Fig. 3c, f and Supplementary Movie 3). Consistent with the notion that each tooth was produced by a single microvillus, the channel tips on the outer edge of the blade exhibited the

same left/right alternation that we observed for the teeth (Fig. 3c, f and Supplementary Movie 3). Moreover, the channels tips at the base of the bristle (Fig. 3d i–iv) exhibited a gradual increase in the length of the respective tooth that is also detectable in light-microscopic analyses (Fig. 3a).

As the channels in the bristle matrix must reflect the respective orientation and dynamics of the microvillar assembly during blade synthesis, these data allow us to propose a first mechanistic model of how individual teeth are engineered during bristle biogenesis (Fig. 3g–k). This model predicts that each microvillus undergoes an extension phase 1 (Fig. 3g–i), during which chitin biosynthesis close to its tip leads to the growth of a new tooth. After chitin synthesis ceases, a disassembly phase 2 (Fig. 3j, k) would leave the characteristic channels in the chitin matrix. Systematic displacement of old microvilli by newly added microvilli at the serrated edge would push the disassembling microvilli to the opposite end of the triangular microvillar assembly over time.

Using the same SBF-SEM strategy, we also investigated and reconstructed the internal structure and channel orientation of larvae sampled during joint and shaft formation (Supplementary Figs. 3, 4 and Supplementary Movies 4–7, respectively). These analyses revealed additional details about changes in microvillar geometries that inform the conformations observed in our light-microscopic analyses (Fig. 2). For instance, our analyses uncovered that the axial microvillus of the shaft originates from a central microvillus in the boss of the joint that grows dramatically in diameter (ac/amv in Supplementary Fig. 3) to finally occupy the majority of the shaft (Supplementary Fig. 4). By contrast, some of the microvilli surrounding the nascent axial microvillus in the boss gradually repositioned to form slim, tapered annular microvilli around the circumference of the axial microvillus at the shaft base (Supplementary Fig. 4).

Taken together, both our light- and electron-microscopic analyses support a tight connection between the different microvillar geometries of the chaetoblast at distinct stages, and the production of micrometric and submicrometric features of the respective bristle. For the production of teeth, the triangular geometry of microvilli, as well as the details of the SBF-SEM-based model, are consistent with a printing mechanism in which new microvilli spawn at one edge of the blade, and extend for four or more "cycles" while depositing chitin close to their apex, before starting to disassemble again, leaving a channel in the solidified chitin matrix.

### Changes in F-actin polymerisation rate impact on feature deposition

Such a printing mechanism would predict that microvillar extension is required for depositing teeth. To test this notion, we decided to use a small molecule approach. Cytochalasins are fungal metabolites that bind to barbed or fast-growing ends of microfilaments and reduce the addition and dissociation of monomeric actin at the barbed end[24]. They have been shown to interfere with microvillar actin polymerisation in intestinal epithelial brush border preparations[25], and were already employed to probe actin-dependent processes in *Platynereis* development[26]. We treated larvae at 40 hpf with 10 μM cytochalasin D, then washed off the drug at 48 hpf and let larvae develop until fixation at 72 hpf. Specimens were imaged using a field emission gun–SEM to assess bristle morphology. Control specimens (treated with 0.1% ethanol) exhibited normal bristle morphology (Fig. 4a), with teeth positioned along the blade (arrow Fig. 4b). By contrast, in cytochalasin D-treated specimens, we observed that the proximal half of the blade (synthesised during treatment) lacked teeth and appeared smooth (Fig. 4c arrow, d). We noted that the chosen time window also had effects on joint morphology, leading to a smaller opening of the joint, and a partial reduction of dentate substructures in the crown (arrowheads Fig. 4b, d). DIC microscopy (Fig. 4e, f and Supplementary Fig. 5) allowed us to quantify these effects: All of the control animals ($n = 20$)

showed teeth and a properly formed boss (Fig. 4e, g). Cytochalasin D-treated larvae ($n = 19$) exhibited a lack of teeth, and a smooth and oval shaft (Fig. 4f, g). Cytochalasin D treatment thus impairs the formation of the teeth and joint features, at a time window (40–48 hpf) when these structures are typically produced. Taken together, our results support that the dynamicity of F-actin is required for the production of micro- and submicrometric bristle features, revealing a critical component of morphogenesis at this largely unexplored size scale.

## Discussion

Our results provide mechanistic and temporal insight into the question of how morphogenesis is driven on the micro- to submicrometric scale, supporting and refining the classical proposal[7] that both the external shape and the internal channels of bristles are consequences of the dynamic modulation of microvillar geometries on the cortex of individual chaetoblasts. These include large microvillar structures, such as the axial microvillus that we find to be regulated in diameter, but also fine microvilli, such as the annular microvilli of the shaft and the microvilli of the forming blade. Our time estimates indicate that the relevant remodelling of the chaetoblast cortex occurs on the level of several hours (large microvilli) to minutes (repetitive details of teeth and diaphragms). A further dynamic aspect are the systematic changes in the relative positioning of blade microvilli, as well as the annular microvilli of the shaft.

We introduce the 3D reconstruction of entire bristles and their internal channel structures using SBF-SEM as an approach to study these dynamic processes. This analysis capitalises on the notion– proposed more than a century ago[6]–that the withdrawal of cellular protrusions leaves channels in the hardened chitin matrix. Spatial reconstructions thus precisely inform about the spatiotemporal changes in the chaetoblast surface geometry over the course of hours. Whereas a recent study has suggested a link between individual microvilli and the formation of teeth, the precise steps of how this would be achieved have remained elusive, and the concept of merging/fusing microvilli has been inferred to explain the retention of a stable blade width despite continuous addition of teeth[9]. Combined with the analysis and manipulation of F-actin geometries, our 3D reconstructions allowed us to derive a distinct mechanistic model for tooth formation: For each tooth, a new microvillus is initiated at the serrated edge of the blade, and extension of F-actin fibres over several cycle lengths is needed for the synthesis of an apical chitin cap. Subsequent disassembly of actin and concomitant retraction is accompanied by displacement of the microvillus to the distal edge of the blade, with a total of 10–12 microvilli present in the entire diameter of the blade at a given time. This model explains both the polarity of the blade and its slender shape, and differs from previous assumptions that teeth would be sculpted as separate structures and fused to the blade[7], or that microvilli would fuse to maintain constant numbers[9].

Moreover, our model also implies the chaetoblast microvilli themselves as relevant sites of tooth synthesis. As outlined before, a long-standing assumption is that microvilli would merely act as moulds for a chitin matrix cast by basal parts of the chaetoblast or adjacent follicle cells[7–9], possibly by shedding of N-acetylglucosamine monomers that spontaneously polymerise in the extracellular space[9]. By contrast, the chaetoblast-specific expression of *cs3*, and the observed similarity of *cs3* crispants with the phenotype obtained by general inhibition of chitin synthases–combined with the localisation of chitin synthases to tips of microvilli in the insect gut[27,28], and the recent validation that chitin synthases act as transmembrane proteins capable of directional synthesis[19]–collectively rather support the alternative model in which the chaetoblast works like the biological correlate of a 3D printer relying on a material jetting principle[10]. Our findings do not contradict the role of the follicle cells in additional chitin synthesis, in

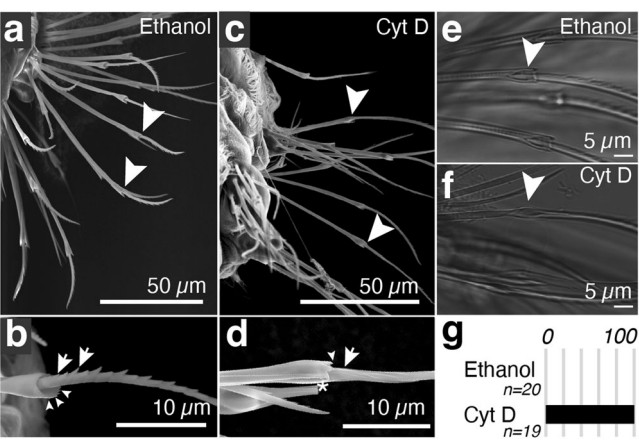

**Fig. 4 | Competing with F-actin extension during blade biosynthesis abrogates teeth formation. a**, **b** SEM Images of larvae that were treated with vehicle (ethanol 0.1%); **c**, **d** corresponding larvae treated with 10 μM cytochalasin D from 40 hpf to 48 hpf. Images were taken at 72 hpf. Arrowheads point to the joint region of the bristle; arrows point to teeth (control) or the corresponding, smooth section on blades synthesised during cytochalasin treatment. **e**, **f** DIC images of bristles present on larvae that were treated with vehicle (**e**) or cytochalasin D (**f**). **g** Percentage of larvae with defects in bristle morphology after treatment with vehicle or cytochalasin D. Source data are provided as a Source Data file.

the secretion of enamel or in the ornamentation of bristles, as has been suggested in nereidid worms and different polychaete groups[9,29]. Our model also reinforces fundamental differences in the processes underlying the formation of fly bristles: Chaetoblasts sculpt extracellular beta-chitin objects, whereas fly bristles are specifically shaped cells coated by an alpha-chitin cuticle[4,28,30]. Moreover, whereas actin bundles also play a role in the shaping of fly bristle cells, only the initial steps rely on surface microvilli, while further growth occurs discontinuously by grafting shorter actin bundles within the cell[31,32].

Whereas we here use the internal channels solely to report the dynamic surface changes of the chaetoblasts, we reason that these channels also have additional functional relevance: Firstly, in the production process, the extension of selected microvilli anchored in the channel may provide a pushing force. Secondly, the channels may also fulfil a mechanics-driven design principle for lightweight structures: to reduce weight without compromising the resistances against bending (by assorting material along the outer dimensions of the cross sections) and against local buckling (by avoiding entirely hollow structures). Bristles thus share design features with far larger, and genetically distant, biological structures, such as feathers and bones[33].

Taken together, the larval bristles of *Platynereis* provide a model for studying the intersection of cell biological processes, material properties and the generation of biological shape on the micro- and submicrometric scale. As an established molecular model system, *Platynereis* may well serve as an experimental reference point to study modifications of a common bristle biogenesis programme over developmental time and in other polychaetes and lophotrochozoans, thereby helping to understand the evolutionary success of these structures.

## Methods
### Animal resources
*Platynereis* worms of PIN and VIO strains were obtained from a laboratory culture maintained at temperatures between 18° and 20 °C in a 16:8-h light-dark (LD) cycle. Larvae were obtained from crosses between mature worms. All experiments and animal handling were carried out in compliance with Austrian ethical guidelines.

### Scanning electron microscopy/SEM
Samples were fixed in 2.5% glutaraldehyde (GA; Agar Scientific, Essex, UK) in seawater overnight and postfixed in 1% osmium tetroxide (Agar Scientific, Essex, UK) in ddH2O. After thoroughly rinsing with water, samples were dehydrated in a 3:1 mixture of 2,2-dimethoxypropane (DMP, Sigma-Aldrich) and water for 10 min, followed by 3x anhydrous acetone for 30 min. each. The larvae were then treated with a 1:1 mixture of anhydrous acetone and hexamethyldisilazane (HMDS, Sigma-Aldrich) and three times in pure HMDS for 1 h each. After several hours of air-drying, the sample was sputtered with 4 nm AuPd 15 mA on a Quorum Q150T Sand and imaged on FEI Quanta 250 field emission gun (FEG)-SEM.

### In situ hybridisation chain reaction (in situ HCR)
Larvae at 48 hpf were dismembered using a glass cover slide against a plastic petri dish in phosphate buffered saline (PBS). Cells were collected and spotted onto poly-lysin-coated cover slides. Cells were allowed to sediment for 10 min. Cells were then fixed in paraformaldehyde (PFA) 4 % for 1 h and finally in methanol at room temperature. Methanol was changed three times and stored in the −20 °C freezer. After 3 days, the sample was rehydrated in increasing quantity of PBS supplemented with Tween-20 0.1% (PTw) in ethanol. The sample was washed twice in PTw for 5 min. then digested in 40 μg/mL of proteinase K (Sigma-Aldrich) for 1 min. at room temperature. The proteolytic reaction was stopped by washing twice with 2 mg/mL glycine in PTw on ice. The sample was then washed in PTw on ice then incubated in 50% probe hybridisation buffer (30% formamide, 5x saline-citrate solution (150 mM NaCl, 15 mM sodium citrate), 9 mM citric acid at pH 6, 0.1 % Tween-20, 50 μg/mL heparin, 1x Denhardt´s solution, 10% dextran sulphate) in PTw for 4 min. at room temperature. The sample was pre-hybridised in 100% probe hybridisation buffer at 37 °C for 1 h. The sample was hybridised overnight at 37 °C in a probe hybridisation buffer containing 4 nM probes that were designed against *chitin synthase 3* of *Platynereis dumerilii* (*cs3*; Genbank ID: KJ405470.1) using the HCR probe generator 3.0 (https://github.com/rwnull/insitu_probe_generator). The sample was washed four times for 15 min. with the probe wash buffer (30% formamide, 5x saline-citrate solution, 9 mM citric acid at pH 6.0, 0.1% Tween-20, 50 μg/mL heparin) at 37 °C. The sample was washed twice for 5 min. in a solution containing saline-citrate solution supplemented with 0.1% Tween-20. The sample was then pre-amplified in amplification buffer (5x saline-citrate solution, 0.1% Tween-20, 10% dextran sulphate) for 30 min. at RT. The initiator was amplified in an amplification buffer containing 60 nM of preheated hairpin h1 and of h2 conjugated to Alexa 546 (Molecular Instruments). The sample was incubated at room temperature overnight. The next day, the sample was washed in PBS four times for a total of 1.5 h, then washed in Tris-buffered NaCl (50 mM Tris pH 7.4 with 150 mM NaCl) for 5 min. The sample was labelled with 50 μg/mL of wheat-germ agglutinin conjugated to Alexa Fluor 488 dye (Thermo Fisher Scientific) in Tris-buffered NaCl, overnight. The next morning, the sample was washed in Tris-buffered NaCl three times and labelled for 20 min with 10 μg/mL 4′,6-diamidino-2-phenylindole (DAPI) (Biotium) in Tris-buffered NaCl. The sample was washed in PBS three times and finally mounted onto a glass slide in VECTASCHILD Antifade Mounting Medium (Vector Laboratories), then imaged through a Plan-Apochromat 63x/1.4 Oil DIC, WD 0.19 mm on a Zeiss Laser Scanning Microscope (LSM) 980 with 32 Channel Gallium Arsenide Module and Airyscan2 detector for super-resolution in multiplex mode with 4x parallel detection.

### DIC imaging of larvae
Larvae were collected at 72 hpf and fixed in PFA 4% in artificial seawater (0.5 M NaCl, 2.5 mM KCl, 10 mM NaHCO3, pH 8). Larvae were spotted on 384 Well Black/Clear Bottom Plate (Thermo Fischer Scientific) and

imaged with Zeiss Observer Z1 inverse microscope mounted with Plan-Apochromat 10x/0.45n RMS and detected with CoolSnapHQ2 camera.

## Cas9-mediated knock-out analyses

Embryo collection and microinjections followed previously established procedures[34,35]. Zygotic microinjection was performed with a solution containing 30 ng/μl Tetramethylrhodamine TRITC dextran (70 kDa), 200 ng/μl *cas9* mRNA and 20 ng/μl synthetic sgRNA. The variable region of *cs3* sgRNA was targeted against the sequence 5′-TC TAATCAGGAGAGGATCGA[TGG]−3′ (underlined sequences represent the associated PAM sequences not included in the sgRNA). For control injections, we used the target sequence 5′-GGGGATGAGTTGGCT-GAGGT[CGG]−3′ of the unrelated *cwo* gene (provided by Federico Scaramuzza and Kristin Tessmar-Raible). Following a phenotypic investigation at 3dpf, single larvae were lysed and used for genomic PCR using the primers 5′-CACTACGTCCAGTATGATCAAG-3′ and 5′-CCATCATGTTGGCCAAGGTGGAG-3′ to amplify a 348 bp fragment encompassing the target site. For restriction tests, these fragments were digested with Sau3AI restriction enzyme and analysed by agarose gel electrophoresis. For sequence analyses, amplicons were subcloned into pJet1.2/blunt vector (Thermo-Fisher Scientific), and individual clones were subjected to Sanger sequencing. Results were compared with the wild-type *cs3* gene model using the CLC Main Workbench software.

## Biochemical isolation of bristles and imaging

Larvae were washed twice using artificial seawater. The larvae were washed in artificial seawater supplemented with 25 mM EGTA, then lysed for 1 h in Triton X-100 1% in the same buffer on ice. We could observe the appearance of a gel containing bristles. The gel was washed twice in 10 mM Tris-HCl, 2.5 mM MgCl₂, 0.5 mM, CaCl₂ pH 7.6. 50 μL were left in the sample at the last wash, and the gel was digested with 5 U of DNase I (Thermo Fischer Scientific) for 2.5 h at 37 °C with 360 rpm agitation. Next, the digest was supplemented with 100 μL of artificial seawater and digested with 200 μL of 2.5 μg/μL Liberase Enzyme Blend (Sigma-Aldrich) for 1 h at 50 °C. The bristles were washed three times in artificial seawater. The isolated bristles were mounted on μ Slide I Luer (Ibidi). The refractive index tomograms of isolated bristles were obtained by Nanolive 3D Cell Explorer, and raw data were deposited at https://doi.org/10.5281/zenodo.10207240.

## Confocal microscopy

The larvae were washed in artificial seawater and then washed in 0.1 M piperazine-*N*,*N*′-bis(2-ethanesulfonic acid)−NaOH (PIPES-NaOH) pH 6.9, 0.5 M NaCl. Larvae were fixed in 0.1 M PIPES-NaOH pH 6.9, 2 mM EGTA, 1 mM MgSO₄, 0.1 mM EDTA and 4% PFA for 40 min. Incubate in glycine 100 mM in PIPES-NaOH 0.1 M pH 6.9 for 40 min. Fixed cells were extracted with Triton X-100 0.5% in 3% bovine serum albumin, glycine 100 mM, 0.1 M PIPES-NaOH pH 6.9, NaN₃ 0.025% incubated overnight. The next morning, the sample was washed in PBS twice and then labelled with 1:1000 phalloidin-iFluor 488 (abcam), 300 nM DAPI, for 1 h. The sample was then washed in PBS three times and mounted onto a glass slide. Finally, the sample was imaged through a Plan-Apochromat 63x/1.4 Oil DIC, WD 0.19 mm on a Zeiss LSM 700. Confocal images were subjected to image restoration by Huygens Deconvolution.

## Sample preparation for serial block face−scanning electron microscopy/SBF-SEM

Larvae at 48 or 64 hpf were isolated using 70 μm meshes, rinsed in 0.1 M PIPES pH 8, NaCl 400 mM, 2 mM CaCl₂, then fixed in PFA 2%, GA 2.5% in the same buffer for 2 h. The sample was left to sediment by gravitation with occasional vibrations. Buffer was exchanged with PFA

2% in the same buffer and shipped. The samples were washed in H₂O and stained with an adapted National Center for Microscopy and Imaging Research protocol[36] aided by microwave application using a Pelco Biowave Pro+ microwave (MW) processing system (Ted Pella, Redding, CA). First, the samples were postfixed in 2% osmium tetroxide and 1.5% potassium ferrocyanide in 0.1 M cacodylate buffer supplemented with 2 mM CaCl₂ in the microwave oven (2 min 100 W, 2 min pause, 2 min 100 W, x3 cycles under vacuum; between the cycles the samples were taken out of the oven for stirring) and then washed x3 times in H₂O (40 secs, 250 W, each). After washing, the samples were incubated in 1% aqueous solution of thiocarbohydrazide (twice, 6 min. of 2 min. MW on/off cycles under vacuum) and washed again (as before). This was followed by incubation in 2% aqueous osmium tetroxide solution (14 min of 2 min 150 W MW on/off cycles under vacuum). Following the second exposure to osmium, the samples were washed (as before) and then incubated in 1% uranyl acetate overnight at 4 °C. The following day, after the samples were washed (as before), the en-bloc Walton's lead aspartate staining was performed. In this step, the samples were incubated in 0.0066 mg/ml lead nitrate in 0.03 M aspartic acid (pH = 5.5) at 60 °C for 30 min, after which the samples were washed (as before). The dehydration was done under MW using increasing concentrations of ethanol: 20%, 50%, 70%, 90%, 96%, three times 100% (40 s each, 250 W without vacuum), and finally placed in ice-cold anhydrous acetone at room temperature for 10 min. The embedding was performed in Durcupan ACM resin (Sigma-Aldrich). First, the samples were infiltrated with Durcupan resin without component C through increasing portions of resin/acetone ratios: 1:3, 1:1 and 3:1, followed by 2 × 100% Durcupan with component C; each step was aided by MW (3 min, 250 W, under vacuum) and finally embedded into Beem embedding capsules (Electron Microscopy Sciences) and baked in a 60 °C oven for 48 h. The blocks were then mounted on aluminium specimen pins (EM Resolutions Ltd, Sheffield, UK) using conductive silver epoxy (CircuitWorks CW2400) and trimmed in a pyramidal shape. Then, the entire surface of the specimen was sputtered with a 5-nm layer of platinum coating (Q150TS coater, Quorum Technologies, Laughton, UK) to improve conductivity and reduce charging during the sectioning process.

## SBF-SEM data acquisition

All SBF-SEM data were acquired using a Quanta 250 Field Emission Gun−SEM microscope (FEI Co., Hillsboro, OR) equipped with a 3View system (Gatan Inc., Pleasanton, CA) using a backscattered electron detector (Gatan Inc., Pleasanton, CA). All the samples were imaged with a beam voltage of 2.5 kV, spot size 3, pressure of 0.08−0.22 Torr and 5 ms dwell time. After imaging, Microscopy Image Browser (MIB)[37] was used to process and align the SBF-SEM image stacks.

## Modelling of bristles

Bristle models were generated in MIB[37]. To speed up the process, the bristle of interest was cropped from the full volume and the structures of interest on each ~10−15 slice were manually segmented. The segmentations were further targeted into DeepMIB[38] to train a U-net[39] for semantic segmentation. After that, the trained U-net was used in DeepMIB to generate the full model of bristles. Finally, the generated model was manually checked and corrected in MIB and visualised in Amira software (Thermo Fisher Scientific).

## Reporting summary

Further information on research design is available in the Nature Portfolio Reporting Summary linked to this article.

## Data availability

Raw data acquired by Nanolive are accessible on Zenodo [https://doi.org/10.5281/zenodo.10207240], and data obtained by SBF-SEM

imaging are accessible on EMPIAR [https://doi.org/10.6019/EMPIAR-11851]. Source data are provided with this paper.

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

## Acknowledgements

The authors are grateful to Graham Warren and Tessmar/Raible lab members for continuous input into this project, and Graham Warren and Jan-Michael Peters for valuable input into the manuscript; the University of Vienna for support of the Marine Facility, and Margaryta Borisova, Andrij Belokurov and Netsanet Getachew for animal supply; the Max Perutz Labs BioOptics Facility for access to advanced light microscopy; the Vienna BioCenter Electron Microscopy Facility for help in sample preparation; E.J. and I.B. acknowledge support from Euro-BioImaging, the Finnish Advanced Microscopy Node, Biocenter Finland, and Helsinki Institute of Life Science. This research has been supported by the Austrian Science Funds (FWF) I2972 (F.R.; doi:10.55776/I2972), F78

(F.R.; doi:10.55776/F78), and Y1093 (J.F. doi: 10.55776/Y1093); For open access purposes, F.R. has applied a CC BY public copyright license to any author accepted manuscript version arising from this submission. Additional support was provided by the Austrian Academy of Sciences (OeAW) Innovation Funds Grant LS19-033 Bio3Dprint (F.R. and C.H.); and the Ministry of Health of the Czech Republic NU22J-08-00062 (J.G.).

## Author contributions

Conceptualisation: K.N.I. and F.R. Investigation: K.N.I., I.B., L.Z.-L., J.G. and L.O. Data curation: K.N.I. and I.B. Methodology: I.B., J.G. and E.J. Funding acquisition: F.R., C.H. and J.F. Supervision: F.R., C.H., E.J. and J.G. Resources: E.J., J.G. and J.F. Visualisation: K.N.I., I.B. and F.R. Writing: F.R., K.N.I. and C.H. Correspondence and requests for materials should be addressed to K.N.I. and F.R.

## Competing interests

The authors declare no competing interests.
