## [Peer Review File · Nature Communications]

Dynamic microvilli sculpt bristles at nanometric scaleREVIEWER COMMENTS

Reviewer #1 (Remarks to the Author):

The manuscript provides novel insights into the formation of chaetae (bristles) in the model annelid species *Platynereis dumerilii*. The researchers combine various microscopic and molecular techniques to develop a model for chaetal pattern formation at a sub-micrometer scale. The scope of the work is impressive, and the text is expertly complemented by excellent figures, including the extended and supplementary material. The supplementary videos are very helpful for visualizing the described processes.

One of the most important results is that chitin is secreted by the chaetoblast itself as opposed to only the follicle cells. As far as I know, this is the first time cas9/CRISPR technology has been used to address questions of chaetal formation. The result is valuable, but it does not exclude the possibility that the follicle cells produce additional chaetal material, only that there is a chaetoblast-specific chitin-synthase gene. This should be discussed. Another very interesting aspect is the temporal sequence. For example, for example the time it takes for a new tooth or a new diaphragm to form.

A lot of the information presented is not exactly new, but the authors have used novel methods to explain underlying molecular mechanisms. For example, the extension and subsequent retraction of microvilli using actin polymerization has long been documented using transmission electron microscopy, but, to my knowledge, the dependency of the chaetal formation on actin polymerization has not been demonstrated by inhibiting it.

What bothered me about the manuscript is the limited credit the authors give to the careful ultrastructural work on chaetogenesis that dates back several decades. Tilic et al. (2023; listed in reference list as 2022, ref. 9) thoroughly documented chaetogenesis in *P. dumerilii* larvae and adults and described the single-microvillus origin of the teeth along the chaetal blade. The work is cited but not in that context. I suspect that both groups worked on the same questions from different angles, but Tilic et al. published their findings first.

Overall, I am impressed by the novel methodological approaches that the authors have taken to address a “classic” question. The work has been beautifully executed, but I think the authors need to state more clearly what is truly new knowledge and acknowledge previous work more explicitly.

Reviewer #2 (Remarks to the Author):

Kyojiro N Ikeda and colleagues present in their manuscript entitled “Dynamic microvilli sculpt bristles at nanometric scale” their results on the cellular mechanisms of bristle-like structure formation in the polychaete worm *Platynereis demerilii*.

They challenge the classical hypothesis that the chitinous extracellular matrix of the bristles is formed by follicle cells adjacent to the actual bristle carrying chaetoblasts. Using histological and genetic data, they indeed show that the chaetoblasts themselves synthesize chitin. Next, they study the dynamics of the

apical plasma membrane of chaetoblasts in forming bundles of microvilli that in turn stepwise sculp the bristle.

The work is exciting; the data are well documented and of high quality. For a more convincing presentation, I suggest addressing the following points:

- The first part of the manuscript deals with the hypothesis of the chitin matrix deriving from the chaetoblasts themselves. This is, honestly, not very surprising as the hypothesis, to me, is very weak as such. This part would be much more interesting if the microvillus phenotype of the cs3 crispants was studied and communicated. Also, I would suggest localizing the chitin synthase protein in the chaetoblasts: are they at the tips of the microvilli? In insects, the chitin synthase complex sits on the tips of membrane protrusions. Is there any analogy? This would certainly strengthen the manuscript, as it would add a molecular information to the cellular mechanism proposed.
- The number of microvilli (page 5: 10) and the number of teeth on bristles (page 4: 21) is not the same at stage I. Guessing that at stages II and III the total numbers equal each other, we, however, need to be informed about this correlation in order to appreciate the proposed progressive mechanism of bristle formation.
- The empty channels (page 6): the authors propose that these channels are left behind by retracting microvilli. A major argument of their model is based on this assumption. However, as microvilli are rather delicate structures, may the fixation method be the reason why we do not see them on SBF-SEM? What about the actin staining? Does it correlate with this retraction observation?
- The cytochalasin experiment: a single concentration was used – why? How was the concentration determined to be good? As actin is an essential component of cells, how can the authors be sure that especially actin problems in the microvilli are the reason of the bristle-less phenotype?
- Bristle/hair formation in insects – also based on a chitinous matrix - namely *Drosophila* has been studied in detail. A few detailed words, hence, in comparison to the polychaete system would add valuable information.

IKEDA ET AL. -- POINT-TO-POINT RESPONSES TO THE REVIEWERS

Reviewer #1

*The manuscript provides novel insights into the formation of chaetae (bristles) in the model annelid species *Platynereis dumerilii*. The researchers combine various microscopic and molecular techniques to develop a model for chaetal pattern formation at a sub-micrometer scale. The scope of the work is impressive, and the text is expertly complemented by excellent figures, including the extended and supplementary material. The supplementary videos are very helpful for visualizing the described processes.*

One of the most important results is that chitin is secreted by the chaetoblast itself as opposed to only the follicle cells. As far as I know, this is the first time cas9/CRISPR technology has been used to address questions of chaetal formation. The result is valuable, but it does not exclude the possibility that the follicle cells produce additional chaetal material, only that there is a chaetoblast-specific chitin-synthase gene. This should be discussed.

ANSWER: We thank the reviewer for their study of our manuscript and their acknowledgement of the overall quality of our work. Concerning this point, the specific expression of *cs3* transcripts in the chaetoblast, combined with the strong phenotype obtained by *cs3* knock-outs, indeed makes a strong case for a central role of the chaetoblast itself in chitin biosynthesis. But as the reviewer points out, we cannot fully rule out chitin synthesis also in the follicle cells. Moreover, secretion of other components (“enamel”) by follicle cells has been discussed in various systems, such as the ornamentation of bristles of the sea mouse *Aphrodite*. This reference is included in the manuscript (Tilic et al 2016). Following the suggestion of this reviewer, we have now modified this section of the discussion (i) to be more precise about the arguments that lead us to our conclusion, but (ii) also make an explicit statement that also in the *Platynereis* system, secretion of chitin from additional sources cannot be fully excluded:

“Moreover, our model also implies the chaetoblast microvilli themselves as relevant sites of tooth synthesis. **As outlined before**, a long-standing assumption is that microvilli would merely act as molds for a chitin matrix cast by basal parts of the chaetoblast or adjacent follicle cells [...], **possibly by shedding of N-acetylglucosamine monomers that spontaneously polymerize in the extracellular space [...]. By contrast, the chaetoblast-specific expression of *cs3*, and the observed similarity of *cs3* crispants with the phenotype obtained by general inhibition of chitin synthases – combined with the localisation of chitin synthases to tips of microvilli in the insect gut [...], and the recent validation that chitin synthases act as transmembrane proteins capable of directional synthesis [...]**– **collectively** rather support the alternative model **in which the chaetoblast works like** the biological correlate of a 3D printer **relying on a material jetting principle [...].**

Our findings do not contradict a role of the follicle cells **in additional chitin synthesis**, in the secretion of enamel or in the ornamentation of bristles, as has been suggested **in nereidid worms** and different polychaete groups [...]

Another very interesting aspect is the temporal sequence. For example, for example the time it takes for a new tooth or a new diaphragm to form.

ANSWER: We thank the reviewer for pointing out this aspect. Indeed, to our knowledge, there has been no attempt to provide temporal details on the predicted surface dynamics, while this information is relevant to understand the nature of the underlying processes .

A lot of the information presented is not exactly new, but the authors have used novel methods to explain underlying molecular mechanisms. For example, the extension and subsequent retraction of microvilli using actin polymerization has long been documented using transmission electron microscopy, but, to my knowledge, the dependency of the chaetal formation on actin polymerization has not been demonstrated by inhibiting it.

ANSWER: As the reviewer mentions, and as we also credit in our study, TEM has been used before to investigate the fine geometry of the chaetoblasts in various systems. But as we argue in the paper, this work has essentially remained descriptive, and experimental work to establish causal connections has not been performed. O'Clair and Cloney, lacking suitable methodology in 1974, concluded: "[it] should be possible to test the proposed hypothesis if a method can be found to alter setal morphology by temporarily modifying the configuration of the chaetoblast during chaetogenesis. The microvilli are supported by highly aligned thin filaments and it might be possible to reversibly alter the shape of the microvilli by briefly interfering with the organization of these filaments." To our knowledge, our work is indeed the first study to explore this modification, and thus establish causality.

What bothered me about the manuscript is the limited credit the authors give to the careful ultrastructural work on chaetogenesis that dates back several decades. Tilic et al. (2023; listed in reference list as 2022, ref. 9) thoroughly documented chaetogenesis in *P. dumerilii* larvae and adults and described the single-microvillus origin of the teeth along the chaetal blade. The work is cited but not in that context. I suspect that both groups worked on the same questions from different angles, but Tilic et al. published their findings first.

Overall, I am impressed by the novel methodological approaches that the authors have taken to address a "classic" question. The work has been beautifully executed, but I think the authors need to state more clearly what is truly new knowledge and acknowledge previous work more explicitly.

ANSWER: We thank the reviewer for their additional remarks. As the reviewer points out, there has been light microscopy and ultrastructural work on a number of polychaetes preceding our work, reaching back more than 100 years. In addition to the reviews by Hausen (*Hydrobiologia*, 2005) and Merz and Woodin (*Integr Comp Biol* 2006), which contain a lot of the relevant references, we cite the experimental work by O'Clair and Cloney (relevant for the concept of dynamic microvilli) as well as Tilic and Bartolomaeus (*Cell Tissue Res* 2023, s.b.) on *Platynereis* and Tilic et al. (*Acta Zoologica*. 2022) on *Aphrodite* in the original manuscript. We have now also added the seminal study by Pruvot (1914) that introduced the principle of a dynamic chaetoblast surface (even though wrongly considering the extensions to be motile cilia).

Moreover, to better point out the existence of ultrastructural descriptions, we now point out more clearly that the reviews cited contain references of prior ultrastructural work so that the readers are directed to this rich body of literature:

“This process yields highly stereotypical geometries, ranging from feather-like structures or hooks to composite structures containing miniature joints (see reviewed in refs. [...] for an extensive review, including ultrastructural work in various polychaete genera).”

“Building on fundamental light microscopy work on the *Nereis* genus that reaches back more than a century [...], electronmicroscopic work in the nereidid polychaete *Nereis vexillosa* revealed specific cells (chaetoblasts) at the base of each bristle that exhibit varying surface geometries matching the synthesized structure [...].”

The reviewer also asks us to more specifically refer to the study of Tilic et al. who have performed an independent analysis of chaetal shapes in *Platynereis*. We include below a longer explanation of shared aspects and major differences, followed by a synopsis of how we have incorporated these points in a compact format in the revised paper.

Indeed, Tilic et al. have recently performed detailed TEM studies on individual specimens of *Platynereis*, including larval stages, in a manner comparable to the cited O'Clair and Cloney work. Overall, their results are in good agreement with the O'Clair and Cloney findings, with the exception that Tilic et al suggest that in *Platynereis*, teeth are correlating with individual microvilli, rather than assemblies of several microvilli (a notion we also support, but based on a very different mechanistic model / s.b.).

However, as the study by Tilic et al. lacks (a) gene expression and functional methodology (knock-out), (b) systematic reconstruction of entire microvillar channels (by serial block-face SEM), and fails to acknowledge the trans-membrane nature of the relevant chitin synthase, their conclusions differ in fundamental points and arrive at divergent interpretations:

(1) Tilic base their interpretation on the model that microvilli serve as a mold, while the material of the bristle is largely secreted by neighbouring follicle cells. The idea of microvilli serving as molds was, to our knowledge, proposed by Pruvot in 1913 (cited in the revised version of the manuscript), and the specific emphasis on the neighbouring follicle cells as source goes back to O'Clair and Cloney 1974. Although Tilic et al call their model a 3D printer like process, it diverges from the way a 3D printer acts; it is rather what we call a “mold/cast” mechanism.

By contrast, our own results (*cs3* localisation, *cs3* knock-out phenotype and its similarity to biochemical interference with all chitin synthases) strongly support the alternative notion that the microvilli themselves are the main producers of chitin and print the bristle, in a fashion comparable to a “material jetting” design of a 3D printer that does not require a mold. Our view agrees with what is proposed in Warren 2015 (with whom we first developed the 3D printer analogy), contrasting with the hypothesis of O'Clair and Cloney 1974.

As to the specific mode of chitin production, Tilic et al argue that the involved cells are "continuously releasing N-acetylglucosamine". They reason that "these molecules continuously polymerize to chitin between the microvilli". In other sections, they speculate that "vesicles [...] transport [...] chaetal material within the follicle cells." and release them into the gap between follicle cells and chaetoblasts.

These views do not align with the notion that chitin synthases are confirmed to be transmembrane proteins that synthesize chitin across the plasmamembrane (cf. the recent structural work by Chen et al. Nature (2022) that we refer to). Moreover, the claim that N-acetylglucosamine monomers spontaneously polymerize in the extracellular space does not agree with the biochemical knowledge that the chitin of bristleworm bristles exhibits parallel chain polarity (so called beta-chitin). This alignment cannot reasonably be explained with a random extracellular polymerization process.

By contrast, the production of beta-chitin is far more plausibly explained by closely spaced chitin synthase proteins residing on the apical chaetoblast membrane that synthesize chitin fibers across the membrane with the same chain polarity. In agreement with this, our first results of anti-Cs3 antibody stainings (see reply to reviewer 2) are fully consistent with the presence of Cs3 on the tip of chaetoblast microvilli, a notion that would be in agreement with a microvillar chitin synthase described in the insect *Manduca sexta* (see below, reviewer 2).

(2) A second fundamental difference to the conclusions by Tilic et al. concerns the mechanistic aspects of in which ways the surface of the chaetoblast changes shape over time. Here, our rigorous analysis of >2000 individual chaetoblast surface geometries gives us a strong basis for recognizing fundamental patterns, which puts us in a different position than Tilic et al, who have analysed, even though at high resolution, only few individuals. This might have contributed to very different views on what happens to microvilli over time.

One of the most fundamental differences we see is that both in the description of the blade and in their interpretation of the biogenesis of the massive axial microvillus of the shaft, Tilic et al. claim that there is "merging" or "fusion" of microvilli. For the blade, they argue, "the constant merging of microvilli maintained a definitive diameter". For the shaft, they claim that "all microvilli merged to form the shaft. The central microvilli fused to form a single large centrally located microvillus. This microvillus then formed the large, central canal [...]"

We arrive at a totally different conclusion: For the blade, our serial sections reveal the precise orientation, beginning and end of the internal channels. showing that there is no fusion, but a constant replacement of microvilli from the serrated face of the blade to the non-serrated face, leading to a slightly oblique orientation of channels, and not inferring any fusion process.

Likewise, for the shaft, our reconstructions very clearly show that the axial microvillus grows in diameter, but not by fusion events. To our knowledge, such a fusion event would also be very difficult to reconcile with our cell biological knowledge of microvilli in other systems.

To accommodate these aspects, but still remain in the compact format of Nature Communications, we have chosen the following procedure:

When introducing the *Nereis vexillosa* work, we now explicitly also mention the study by Tilic et al., performed on individual specimens, as supportive of the general principles:

“Consistent with the role of microvilli in polychaete bristle biogenesis, classical ultrastructural work has found cellular protrusions to be associated with bristles in the annelid *Nereis vexillosa* [...], **and recent analyses have confirmed largely similar geometries in chaetoblasts of *Platynereis* individuals fixed at different stages of development [...].** To systematically probe for a correlation of microvillar geometry at the cell cortex of *Platynereis* chaetoblasts and the biogenesis of diagnostic bristle features, we performed a systematic analysis of chaetoblast microvilli at distinct times of development [...].”

To contrast the conceptual differences, we have modified the respective sections in the discussion as follows:

“**Whereas a recent study has suggested a link among individual microvilli and the formation of teeth, the precise steps of how this would be achieved have remained elusive, and the concept of merging/fusing microvilli has been inferred to explain the retention of a stable blade width despite continuous addition of teeth [...].** Combined with the analysis and manipulation of F-actin geometries, **our 3D reconstructions allowed us to derive a distinct mechanistic model for tooth formation:**
[...]

This model explains both the polarity of the blade and its slender shape, and differs from previous assumptions that teeth would be sculpted as separate structures and fused to the blade [...], **or that microvilli would fuse to maintain constant numbers [...].**

Moreover, our model also implies the chaetoblast microvilli themselves as relevant sites of tooth synthesis. **As outlined before,** a long-standing assumption is that microvilli would merely act as molds for a chitin matrix cast by basal parts of the chaetoblast or adjacent follicle cells [...], **possibly by shedding of N-acetylglucosamine monomers that spontaneously polymerize in the extracellular space [...].** **By contrast, the chaetoblast-specific expression of *cs3*, and the observed similarity of *cs3* crispants with the phenotype obtained by general inhibition of chitin synthases – combined with the localisation of chitin synthases to tips of microvilli in the insect gut [...], and the recent validation that chitin synthases act as transmembrane proteins capable of directional synthesis [...]**– collectively rather support the alternative model **in which the chaetoblast works like the biological correlate of a 3D printer relying on a material jetting principle [...].**”

We believe that these changes are both faithful to the work performed by Tilic et al., and at the same time highlight more clearly major conceptual insights we have achieved by our study.

Reviewer #2:

Kyojiro N Ikeda and colleagues present in their manuscript entitled "Dynamic microvilli sculpt bristles at nanometric scale" their results on the cellular mechanisms of bristle-like structure formation in the polychaete worm *Platynereis demerilii*.

They challenge the classical hypothesis that the chitinous extracellular matrix of the bristles is formed by follicle cells adjacent to the actual bristle carrying chaetoblasts. Using histological and genetic data, they indeed show that the chaetoblasts themselves synthesize chitin. Next, they study the dynamics of the apical plasma membrane of chaetoblasts in forming bundles of microvilli that in turn stepwise sculp the bristle.

The work is exciting; the data are well documented and of high quality. For a more convincing presentation, I suggest addressing the following points:

- The first part of the manuscript deals with the hypothesis of the chitin matrix deriving from the chaetoblasts themselves. This is, honestly, not very surprising as the hypothesis, to me, is very weak as such. This part would be much more interesting if the microvillus phenotype of the *cs3* crispants was studied and communicated.

ANSWER: We thank the reviewer for their appreciative comments on the quality and relevance of our work.

We politely disagree on the point that the contribution of the chaetoblast to the bristle matrix is not surprising. Indeed, the highly influential paper by O'Clair and Cloney (1974) proposed that "[t]he follicular cells secrete the precursors of setal material. These molecules diffuse onto the apical plasmalemma of the chaetoblast.", where they are then proposed to polymerize and harden. Numerous studies since then have followed this train of thought, including the recent study by Tilic et al., and electron-dense vesicles close to the surface of follicle cells are interpreted to constitute secreted chaetal material. While they might well constitute part of the enamel layer, our work makes a strong point that the bulk of chitin is likely contributed by the chaetoblast itself.

However, we thank the reviewer for raising an interesting conceptual question, namely if the formation of the microvilli themselves depends on the proper production of chitin. To address this request, we have imaged the status of the microvilli in *cs3* crispants and upon treatment with the chitin synthase inhibitor NikZ already introduced in the manuscript. We find that in both crispants and drug-treated larvae, microvilli do form, but their morphology is divergent from the canonical patterns that we described in Fig 2 b-g. This implies that the formation of microvilli is principally independent of chitin synthesis, but that the fine geometry requires some feedback from the formed bristle material. The Fig. 2 in the manuscript, and these supplementary figures are referred to towards the end of the results section in which the regular actin geometries are described:

"To test if the formation of microvilli depended on chitin synthesis, we assessed actin geometries after NikZ-mediated inhibition of chitin synthesis and in bristle-less *cs3* crispants. In both cases, microvilli were consistently present, even though the fine morphology of the microvillar assembly exhibited abnormalities (Supplementary Fig. 1, Supplementary Fig. 2). Taken together, our analyses support the notion that stereotypic

arrangements of a dynamic microvillar programme are correlated with individual steps of Platynereis bristle formation.”

Also, I would suggest localizing the chitin synthase protein in the chaetoblasts: are they at the tips of the microvilli? In insects, the chitin synthase complex sits on the tips of membrane protrusions. Is there any analogy? This would certainly strengthen the manuscript, as it would add a molecular information to the cellular mechanism proposed.

ANSWER: We thank the reviewer for this suggestion. As the reviewer alludes to, there is a highly interesting association of chitin synthase with the microvillar surface of columnar cells in the insect gut (which secretes a beta-chitin structure called the peritrophic membrane). In combination with our model, this localisation is very plausible, as it would place chitin synthase precisely at a position where it could be involved in tooth formation.

Unfortunately, to validate such a localisation does not appear to be a straightforward task. We note that also in the insect field, the most reliable work results from one species – *Manduca sexta* – where Merzendorfer and colleagues used a polyclonal serum against a fragment of chitin synthase to stain the surface of columnar cells (Zimoch, L., and Merzendorfer, H. (2002). *Cell Tissue Res.* 308, 287–297. 10.1007/s00441-002-0546-7.). 20 years past that study, the lack of localised chitin synthases in other systems – and notorious difficulties to express functional chitin synthase protein from various systems – indicates that these molecules are not easy to work with.

Nonetheless, we have used the entire timeframe provided for the revision work to attempt analogous experiments in the *P. dumerilii* system. We obtained a custom-made and affinity-purified rabbit polyclonal antibody raised against a Cs3-specific peptide, and tried analyses in various conditions on Platynereis larvae. As shown in the Figure provided below, which analyses the surface of a chaetoblast exhibiting a stage II geometry, we detect an increase in labelling towards the apical tip of the microvilli (Figure). When assessing other stages, however, we note that the antibody reactivity is reduced where chitin is already deposited, making it technically very challenging to investigate the localisation during blade synthesis, where a localisation would be most informative for the focus of the paper. We strongly suspect that in those chaetoblasts, where the microvilli are more buried in the hardening chitin matrix, the Cs3 epitope may be less accessible, requiring additional refinement of the experimental approach – such as immunogold labelling, which is far from trivial to establish in a short time-frame. Adding to this, we have so far not yet recovered reproductive *cs3* mutant carriers, which has prevented us from expanding and using such a mutant strain for proving that the detected signal is for sure caused by endogenous Cs3.

In conclusion, the data we have been able to obtain in the revision time are fully consistent with the notion that Cs3 protein indeed localizes to the apical tips of chaetoblast microvilli, but it seems that significantly more time will be needed to bring those data to the same quality as the remainder of the paper. If the reviewer wants the preliminary data to be added as a supplementary information, we are happy to do so. Else, we would prefer to characterize the protein better with additional experiments that were beyond the timeframe provided.

In either case, we have added references to two reviews that explain the localisation of chitin synthases in the insect gut to the section of the discussion that explains the plausibility of our model:

“[...] [T]he chaetoblast-specific expression of *cs3*, and the observed similarity of *cs3* crispants with the phenotype obtained by general inhibition of chitin synthases – **combined with the localisation of chitin synthases to tips of microvilli in the insect gut [28,29]**, and the recent validation that chitin synthases act as transmembrane proteins capable of directional synthesis [...]– **collectively** rather support the alternative model in which the chaetoblast works like the biological correlate of a 3D printer relying on a material jetting principle.

[...]

28. Zhu, K.Y., Merzendorfer, H., Zhang, W., Zhang, J., and Muthukrishnan, S. (2016). Biosynthesis, Turnover, and Functions of Chitin in Insects. *Annu. Rev. Entomol.* 61, 177–196. 10.1146/annurev-ento-010715-023933.

29. Moussian, B. (2013). The apical plasma membrane of chitin - synthesizing epithelia. *Insect Sci.* 20, 139–146. 10.1111/j.1744-7917.2012.01549.x.”

Preliminary results from the immunolocalisation of *Platynereis* Chitin synthase 3 (*Cs3*). (a) Reactivity of affinity-purified polyclonal anti-*Cs3* antiserum in a stage II chaetoblast; (b) phalloidin labeling of the microvillar F-actin using phalloidin (cf. Fig. 2e of the manuscript) (c) Merged image showing anti-*Cs3* reactivity in green and F-actin signal in magenta. Anti-*Cs3* signal (green arrows) appears to be strongest close to microvillar tips (white arrowheads). Size marker: 2 μ m

- The number of microvilli (page 5: 10) and the number of teeth on bristles (page 4: 21) is not the same at stage I. Guessing that at stages II and III the total numbers equal each other, we, however, need to be informed about this correlation in order to appreciate the proposed progressive mechanism of bristle formation.

ANSWER: We thank the reviewer for pointing out this potential source of misunderstanding. In fact, the mismatch between the rather constant ~10 microvilli [at any given time] in stage I chaetoblasts and the total number of 21 teeth [generated over the 12 hours of blade biosynthesis] is precisely why we looked for a possible mechanism – which made us discover

the gradual displacement of microvilli from the serrated to the non-serrated edge documented in Fig. 3c.

We have now revised the corresponding text in order to make clear that this is no contradiction, but rather a question of the production mechanism:

“As the formation of a blade requires around 12 hours and **during that time produces a total of 21 teeth (s.b.)**, a new tooth is, on average, initiated every ~35 minutes.”

[...]

“Light-microscopic analyses suggested that individual microvilli in stage I arrangements could be associated with the formation of individual teeth (Fig. 3a). To examine the details of this potential correlation in more detail, and **gain insight into the question how a rather constant set of ~10 microvilli produces twice as many teeth**, we turned to serial block-face scanning electron microscopy (SBF-SEM).”

- The empty channels (page 6): the authors propose that these channels are left behind by retracting microvilli. A major argument of their model is based on this assumption. However, as microvilli are rather delicate structures, may the fixation method be the reason why we do not see them on SBF-SEM? What about the actin staining? Does it correlate with this retraction observation?

ANSWER: The idea that channels are left by retracting protrusions is not our proposal, and it was not our intention to insinuate this. In fact, this concept was, to our knowledge, already proposed by Pruvôt in 1914. In 1974, O’Clair and Cloney then managed to image longitudinal sections of chaetoblasts using TEM, providing detailed views of microvilli and adjacent channels during bristle formation. Their work is fully consistent with the concept of retraction. For instance, channel sizes correlate with the shape of the microvillar tip: A very evident example are the ancillary microvilli of the shaft that have a tapered shape with a pointed tip. These leave very narrow channels (“cortical channels” in O’Clair and Cloney’s Fig. 13). This width cannot be explained with a rapid collapse during fixation (as it should then be much wider), yet matches very well the expectation one would have if the pointed microvillus retracts slowly within a hardening matrix. At the other end of the size spectrum, the very broad axial microvillus of the shaft is known to leave behind a wide central channel with regularly spaced diaphragms. This axial microvillus can be reliably stained by phalloidin, and retains similar lengths throughout the elongation of the shaft. This is fully consistent with periodic retractions, as this microvillus would else be as long as the shaft itself. Finally, the fact that thin, long microvilli like the ancillary microvilli (lateral in Fig. 2g) are maintained after fixation, and are longer than the central axial microvillus, is another argument against lability of the microvilli in our system. These different lengths actually also align perfectly with the TEM data of O’Clair and Cloney in the different species they investigated.

While the concept of the channels as left-overs of retracted microvilli is thus broadly accepted, the way in which we reconstruct these channels and use them to infer the detailed changes in surface geometries is novel, and allows us to correct views such as the idea that channels merge / fuse. As we needed to reconstruct entire bristles, our SBF-SEM imaging operated at a

resolution that did not allow us to properly distinguish empty and filled channels. This is the reason why we spent the effort on categorizing phalloidin stainings (Fig. 2) and matching them with the geometries of the channels.

As the reviewer points out, our model is thus based on the combined observation of the channels and microvilli membrane from the SBF-SEM, and from the phalloidin staining of the F-actin contained in the microvilli. The most direct evidence for retraction is given in the analysis of the blade synthesis mechanism: The consistent shape detected by phalloidin stainings is a triangular arrangement with a left and a right slope (Fig. 2b,c). Correlation with chitin morphology in Fig. 3a clearly shows how one slope of this arrangement fits into the serrated edge of the bristle, and tightly associates with the teeth (arrows), while the other slope does not contact teeth any more. The microvilli that are on the non-serrated side of the blade are thus in fact shorter, consistent with the concept that they are in the process of disassembly. As is evident from our SBF-SEM data (Fig. 3c), channels terminate at the non-serrated edge, and individual cross-sections by O'Clair and Cloney (Fig. 4 of their study) also show that close to that termination point, channels on the non-serrated side are empty, suggesting complete retraction.

Following the comment and recommendations of the reviewer, we have made the following changes in the text (i) to more clearly point out the historic origin of the concept of the channels, and (ii) to clarify better how the actin staining correlates with the retraction observation:

“The inside of the SBF-SEM-based blade model is characterized by longitudinal channels. According to early work in nereidid polychaetes [...], only channels close to the chaetoblast contain the microvilli observed in the aforementioned analysis, whereas the distal channels result from the disassembly of microvilli after the solidification of the chitin matrix (Fig. 3c, Supplementary Video 2).”

[...]

We introduce the 3D reconstruction of entire bristles and their internal channel structures using SBF-SEM as an approach to study these dynamic processes. This analysis capitalizes on the notion – proposed more than a century ago [...] – that withdrawal of cellular protrusions leaves channels in the hardened chitin matrix.

- The cytochalasin experiment: a single concentration was used – why? How was the concentration determined to be good? As actin is an essential component of cells, how can the authors be sure that especially actin problems in the microvilli are the reason of the bristle-less phenotype?

ANSWER: We thank the reviewer for raising this issue. Indeed, we reasoned that too high concentrations of cytochalasin D might be toxic, and had therefore already performed a titration of the compound. We have now introduced a new Supplementary Fig. 5, which illustrates the effect of the cytochalasin D treatment at 40, 20, and 10 μ M concentration. Briefly, we observe that larvae treated for the respective time frame with 40 μ M cytochalasin D retain a normal shape, but bristles do not protrude through the cuticle at the regular time point, and are instead found stuck in the body of the animal. Presumably, this is because the microvillar programme

does not fully recover after such a strong treatment. At 20 μM , we have a similar phenotype to what we have described in the main Fig 4 (which showed a 10 μM treatment).

Taken together, for our SEM imaging, we selected the lowest concentration of cytochalasin D that produced a phenotype, reducing the risk of unspecific effects. In these low concentration treatments and in this brief window of treatment (40-48 hpf), we observe that the overall morphology of the larvae is unaffected.

Supplementary Figure 5 also documents that for all of the used concentrations, larvae form other fine structures such as the motile cilia of the prototroch (used for propagation in the water column) that are microtubule-based. We reason that this demonstrates well a lack of unspecific toxicity of the drug at the employed concentration.

- Bristle/hair formation in insects – also based on a chitinous matrix - namely *Drosophila* has been studied in detail. A few detailed words, hence, in comparison to the polychaete system would add valuable information.

ANSWER: We thank the reviewer for the suggestion. While we had briefly touched on the fundamental differences in the original version, they had indeed not been spelled out. To make the uniqueness of the bristleworm bristles clearer, we have now introduced a short statement in the discussion that highlights the major differences to fly bristles, and also provides some key references for readers to follow up on the topic:

Our model also reinforces fundamental differences to the processes underlying the formation of fly bristles: Chaetoblasts sculpt extracellular beta-chitin objects, whereas fly bristles are specifically shaped cells coated by an alpha-chitin cuticle [6,29,31]. Moreover, whereas actin bundles also play a role in the shaping of fly bristle cells, only the initial steps rely on surface microvilli, while further growth occurs discontinuously by grafting shorter actin bundles within the cell [32,33]

6. Hausen, H. (2005). Chaetae and chaetogenesis in polychaetes (Annelida). *Hydrobiologia* 535–536, 37–52. [10.1007/s10750-004-1836-8](https://doi.org/10.1007/s10750-004-1836-8).

29. Moussian, B. (2013). The apical plasma membrane of chitin - synthesizing epithelia. *Insect Sci.* 20, 139–146. [10.1111/j.1744-7917.2012.01549.x](https://doi.org/10.1111/j.1744-7917.2012.01549.x).

31. Lotmar, W., and Picken, L.E.R. (1950). A new crystallographic modification of chitin and its distribution. *Experientia* 6, 58–59. [10.1007/bf02174818](https://doi.org/10.1007/bf02174818).

32. Tilney, L.G., Connelly, P.S., and Guild, G.M. (2004). Microvilli appear to represent the first step in actin bundle formation in *Drosophila* bristles. *J. Cell Sci.* 117, 3531–3538. [10.1242/jcs.01215](https://doi.org/10.1242/jcs.01215).

33. Tilney, L.G., and DeRosier, D.J. (2005). How to make a curved *Drosophila* bristle using straight actin bundles. *Proc. Natl. Acad. Sci.* 102, 18785–18792. [10.1073/pnas.0509437102](https://doi.org/10.1073/pnas.0509437102).

REVIEWERS' COMMENTS

Reviewer #1 (Remarks to the Author):

Thank you for providing thorough answers to all of my queries. In my opinion, the manuscript is now ready for publication.

Reviewer #2 (Remarks to the Author):

The authors responded to my questions highly satisfactorily. They have added experiments that put the work on even a higher level than before. However, I would have wished that they develop their argument based on classical work on chitin deposition. Firstly, I appreciate science history, secondly, the model of O'Clair and Cloney, as argued by the authors, is pretty naive; therefore, refuting a naive model is not a heroic deed on its own.

The authors are asking whether I would wish to see the Cs3 localisation data. This is very very polite. But this is their manuscript, and I am very satisfied with their experiments presented in their answer, they (or the editor) should, hence, decide to add it or not.

Ikeda et al.

Response to Reviewers

Reviewer #1 (Remarks to the Author):

Thank you for providing thorough answers to all of my queries. In my opinion, the manuscript is now ready for publication.

Reviewer #2 (Remarks to the Author):

The authors responded to my questions highly satisfactorily. They have added experiments that put the work on even a higher level than before. However, I would have wished that they develop their argument based on classical work on chitin deposition. Firstly, I appreciate science history, secondly, the model of O'Clair and Cloney, as argued by the authors, is pretty naive; therefore, refuting a naive model is not a heroic deed on its own. The authors are asking whether I would wish to see the Cs3 localisation data. This is very very polite. But this is their manuscript, and I am very satisfied with their experiments presented in their answer, they (or the editor) should, hence, decide to add it or not.

We thank both reviewers for their constructive comments and for making the review process an enjoyable experience. Following the suggestion of the editor, we have not included the preliminary anti-Cs3 staining in the supplementary information.